# Numerical Analysis of Factors Influencing the Ground Surface Settlement above a Cavity

**DOI:** 10.3390/ma15238301

**Published:** 2022-11-22

**Authors:** Kangil Lee, Junhee Nam, Jeongjun Park, Gigwon Hong

**Affiliations:** 1Department of Construction and Environmental Engineering, Daejin University, 1007 Hoguk-ro, Pocheon-si 11159, Republic of Korea; 2Incheon Disaster Prevention Research Center, Incheon National University, 119 Academy-ro, Yeonsu-gu, Incheon 22012, Republic of Korea; 3Department of Civil Engineering, Halla University, 28 Halladae-gil, Wonju-si 26404, Republic of Korea

**Keywords:** finite element analysis, cavity, influence factor, surface settlement

## Abstract

In this study, ground stability was evaluated through vertical displacement distribution and surface settlement results. In particular, a finite element analysis was conducted considering various factors (namely, cavity type and area, traffic load, pavement thickness, and elastic modulus) that affect a road above a cavity. The aim of this study was to evaluate the effect of pavement layer and traffic load condition on surface settlement according to the cavity shape. Field measurement results were analyzed and compared with the results of previous studies to verify the reliability of the numerical analysis method applied herein. After performing the numerical analysis using the verified method, ground stability was evaluated by analyzing the underground mechanical behavior of a road above a cavity. To this end, the correlations among the vertical displacement distribution, surface settlement, and influencing factors obtained from the analysis results were analyzed. In the numerical analysis, the ground was simulated with a hardening soil model based on the elastoplastic theory. This mechanical soil model can accurately reproduce the behavior of actual ground and can closely represent the mechanical behavior of the soil surrounding a cavity according to the cavity generation. In addition, the elapsed time was not considered when applying a load on the pavement layer, and a uniformly distributed load was applied. Consequently, it was found that, with increasing cavity area and traffic load and decreasing pavement thickness and elastic modulus, the vertical displacement and maximum surface settlement above the cavity increased, and the reduction in ground stability was greater. Furthermore, the reduction in ground stability was greater when the cavity was rectangular than when it was circular.

## 1. Introduction

Recently, the occurrence and scale of phenomena defined as road and ground subsidence in downtown areas have gradually increased. This is considered a major safety hazard because there are concerns about the safety and psychological anxiety of the citizens. With the enforcement of the Special Act on Underground Safety Management [1] in South Korea, ground subsidence (Article 35, Paragraph 1 of the Enforcement Decree) has been established to occur in the presence of areas ≥1 m^2^ or depths ≥1 m or if death, disappearance, or injury occurred to people due to ground subsidence. Frequently, ground and road subsidence in South Korea are caused by human activities. It has been revealed that ground subsidence mostly occurring in downtown areas is caused by cavities resulting from old sewage pipes buried underground [2,3,4,5,6,7,8,9]. According to the occurrence criteria of the Special Act on Underground Safety Management [1], the number of occurrences (Figure 1) of ground and road subsidence (Figure 1) has been constantly increasing since 2014. A survey in 2018 found that the number increased approximately five times, and the increasing trend continued until recently [10,11].

With this background, various studies were conducted worldwide using indoor model experiments and large-scale field experiments to investigate the occurrence mechanisms of ground and road subsidence and the causes of cavity occurrence [11,12,13,14,15,16,17,18,19,20,21,22,23]. Ref. [14] analyzed the expansion mechanism of cavities and relaxation areas in the ground and suggested four major causes. Ref. [15] analyzed the relaxation mechanisms of the surrounding ground and cavity expansion by simulating repetitive sewage runoffs from drainpipes through an indoor model experiment using standard sand in Japan. Ref. [16] used an X-ray CT scanner for the 3D visualization and evaluation of cavity occurrence and failure mechanisms via sediment runoff and water inflow due to cracks in old sewer pipes. Ref. [20] explored the effects of elevation, slope, drainage density, groundwater reduction, and distance from neighboring rivers on ground subsidence. The authors focused on a ground subsidence case in Iran and used a mechanical learning algorithm to evaluate the importance of each influencing factor by considering its effects. Ref. [23] extracted the influencing factors such as wetness index, slope, vegetation index, rock type, groundwater, elevation, and land use by focusing on ground subsidence prediction in the Middle East and Southern Iran. Subsequently, the authors compared and analyzed the predicted sensitivities by applying various machine learning models, such as RF, SVM, and EBF. Meanwhile, the causes of ground subsidence were analyzed using indoor model experiments in South Korea. In particular, various ground conditions, such as ground water level, relative density and compaction, and damage location in water and sewer pipes, were considered. In addition, factors affecting the cavity ground stability were considered [6,11,17,19].

However, clearly identifying the occurrence of cavities and their expansion mechanisms in the ground is difficult. This is because not only the accurate investigation of cavities in the ground is challenging but also the influence of the influencing factors on cavitation is highly diverse. Moreover, although many experimental studies on ground subsidence have been conducted, rigorous analyses of the causes are limited because the tests have been performed under limited damaged sewer pipe conditions. In addition, ground subsidence and collapse caused by underground cavities are due to the physical damage of the ground and underground structures and the engineering characteristics of the ground. Thus, their experimental description is difficult. Therefore, various numerical analyses have been conducted considering many different influencing factors to examine the ground subsidence occurrence mechanism [5,7,24,25,26,27,28,29,30,31,32,33].

Ref. [24] suggested a stability evaluation and stability charts for ground subsidence by conducting 2D finite element analysis (FEA) using the shear strength reduction method in areas with limestone cavities. Ref. [25] performed a numerical analysis to analyze the correlation between fractures in the upper cavity and surface subsidence by considering various cavity types, ground conditions, and influencing factors based on the Hoek–Brown failure criterion. Furthermore, various physical characteristics of the generation and expansion of cavities have been presented through a topographical mechanism analysis of sinkhole and ground subsidence cases [34,35,36]. Ref. [33] analyzed the causes of natural and anthropogenic sinkholes and the magnitude of the damages by building a database of ground subsidence and sinkhole cases in chronological order and suggested environmental management measures and recommendations. Meanwhile, ref. [27] analyzed the conditions with the highest probability of road collapse due to underground cavities within the scope of influencing factors by referring to the cavity management rating system established in Seoul (South Korea). Ref. [5] performed FEA considering the magnitude of sewage pipe damage and cover height and then simulated the cavities and evaluated the magnitude of the cavities and the relaxation areas via the gap ratio and shear stress reduction ratio. Ref. [32] conducted a falling weight deflectometer (FWD) inverse analysis to evaluate the effect of the underground cavity characteristics on the bearing capacity reduction of the asphalt pavement. Furthermore, the authors identified a correlation between the elastic modulus of the asphalt pavement and the cavity condition by calculating the remaining life of the road asphalt pavement in which a cavity existed in relation to the decrease in bearing capacity. Ref. [7] evaluated the mechanical behavior of cavity and relaxation areas using the gap ratio and shear stress reduction ratio by performing FEA considering the factors affecting the underground cavity, such as cover height, damage magnitude, and groundwater level. There are also studies conducted analytically and experimentally on the effects of ground subsidence exerted by tunnels. In a representative study [37,38], the effects of subsidence and cavity were analyzed in the upper ground of the tunnel according to the conditions of ground and tunnel support.

However, the existing numerical analyses have mainly considered influencing factors such as cavity area, pavement layer, and surcharge. Few studies have analyzed the ground behavior according to traffic load and pavement conditions by simultaneously considering the cavity area and shape that directly affect the ground and road subsidence as influencing factors. Moreover, studies that numerically evaluated ground stability according to the change in the stress state of the ground with a cavity considering the above-mentioned influencing factors are also insufficient. Hence, it is believed that a new database with a large amount of data related to underground cavities and their features should be built and analyzed in detail to identify criteria for underground cavity prediction through the analysis of major influencing factors.

The aim of this study was to evaluate the effect of pavement layer and traffic load condition on surface settlement according to the cavity shape. Therefore, the correlation between each influencing factor and the surface settlement of ground above a cavity was evaluated by numerical analysis. In order to verify the reliability of the numerical analysis results, the field measurement results of the ground above a cavity were compared with the numerical analysis results (surface settlement). In addition, factors affecting the ground above a cavity (cavity area and type, change in the elastic modulus considering pavement thickness and cracks, and traffic load) were determined. This was achieved through a verified numerical analysis model and the surface settlement results based on the comparison with results of existing studies [7,29,30,32] and field measurements. Furthermore, the ground mechanical behavior and stability were evaluated using the vertical displacement distribution and a correlation analysis of each influencing factor and the maximum surface settlement based on the cavity characteristics affecting the ground.

## 2. Materials and Methods

### 2.1. Verification of the Numerical Analysis Method

In this study, FEA was performed using PLAXIS 2D (ver.21.01., release: April 2021), a general-purpose application that can apply mechanical models for various soil types and analyze the soil mechanical behavior in detail. The mechanical soil model, boundary conditions, soil parameters, and basic numerical analysis method applied to this numerical analysis were determined by referring to previous studies [7,30]. Based on previous studies, a hardening soil model based on the elastoplastic theory [39,40] was applied. This mechanical soil model can accurately reproduce the behavior of actual ground and can closely represent the mechanical behavior of the surrounding soil according to cavity generation. The soil parameters applied to the FEA are listed in Table 1. In general, the parameter E_50_ applied to the hardening soil model is the elastic modulus with increasing load. That is, the elastic modulus according to the confining pressure has a value of 50% of the shear strength in the stress–strain curve. In addition, E_ur_ is the elastic modulus for unloading and reloading. In the equation, the coefficient corresponding to the confining pressure of E_50_ and E_ur_ is 0.5, which is generally used in sandy soil. The values converted and applied to the hardening soil model for general sandy soil materials [41] and details of the hardening soil model (Figure 2) can be found in the literature [7,30,39]. Furthermore, a numerical analysis was performed to confirm the mechanical behavior results caused by applying a traffic load as a uniformly distributed load on the asphalt pavement above the modeled ground simultaneously with stress release for simulating the cavity in the modeled ground. In addition, the elapsed time was not considered for the load on the pavement layer, and the uniformly distributed load was applied to the pavement layer. The interface model between the pavement layer and the ground element was simulated in the rigid (R_inter_ = 1.0) state. The pore water pressure due to groundwater infiltration was not considered in this study.

To verify the reliability of the numerical analysis performed herein, FEA methods for the ground with a cavity that were used previously [7,30] and field measurement results of underground cavities [29] were utilized. The field measurement results herein [29] refer to the surface settlement in one underground cavity site, measured by a ground penetrating radar (GPR) survey and an FWD test. The field experiments are summarized as follows. An FWD sensor was installed in the upper pavement layer of the underground cavity at a certain distance from the center of the cavity, and the deflection of the pavement layer by the FWD test was applied as a field test result. In addition, the amount of deflection was corrected to a temperature of 20 °C in consideration of the temperature environment of the pavement layer.

Figure 3 shows the numerical analysis model considering the pavement thickness, cavity area, and top cover height of the cavity at the field measurement point based on the cavity and field conditions surveyed via GPR. The boundary conditions of the numerical analysis model refer to previous studies [7,30]; the displacement was fixed on the x-axis on the side and on the y-axis at the bottom, as shown in Figure 3. To minimize the influence on the side wall of the modeled ground, the modeled ground width was established as 10 m, and rollers were applied to each side to fix the displacement in the horizontal direction for the side and in the vertical direction for the bottom. Moreover, the cavity in the modeled ground was subjected to stress release simultaneously with the load application to simulate the ground above a cavity and to confirm the soil mechanical behavior results. The numerical analysis result has the value of the ultimate limit state.

As described above, the maximum surface settlement on the top of the pavement layer at the point of cavity occurrence (0 mm) was measured and compared with values measured at different points to verify the numerical analysis method. The field measurement and FEA results were compared for surface settlements corresponding to 200, 300, 450, 600, 900, and 1500 mm at points gradually moving away from the center.

The verification showed that the maximum settlement occurred on the surface at the modeled ground center (the uppermost pavement layer from the point where the cavity occurred), showing the same trend as that of the field measurement results, as shown in Figure 4. This means that it can greatly affect the road use, because the bearing capacity of the pavement layer is greatly reduced due to the continuous traffic load. It was found that the surface settlement decreased as the distance from the center of the modeled ground increased. In other words, it was confirmed that the surface settlement and trends were similar. Therefore, the results of the numerical analysis model established with reference to the methods reported in previous studies showed values similar to the field measurement results. This suggests that the numerical analysis method applied in this study can sufficiently simulate the mechanical behavior of ground above a cavity.

### 2.2. Numerical Analysis Considering the Factors Influencing Cavity Occurrence 

Cavity type and area, pavement thickness, elastic modulus, and traffic load were selected and applied as influencing factors in the verified numerical analysis. Rectangular and circular cavity shapes were considered. The cavity areas were set to 0.79, 1.77, and 3.14 m^2^ to consider both small and large cavities. Furthermore, the top cover height of the cavity was set to 1 m to meet the general burial depth standard for old sewage pipes. The pavement thickness was set to a general value of 0.3 m, and the elastic modulus was set to 3000 MPa. Moreover, the pavement thickness was set to 0.2 m and 0.1 m and the elastic modulus was set to 500 MPa considering cracks. The pavement parameters are summarized in Table 2. The traffic load was applied in the range of 12.7–38.1 kN/m^2^ considering the standard truck load at domestic construction sites and the loading weight. The analysis model conditions were the same as in the model verification. Table 3 and Figure 5 summarize the numerical analysis cases and models. 

## 3. Results and Discussion

### 3.1. Vertical Displacement Distribution

Figure 6 shows the vertical displacement distribution result as a distribution diagram, with the displacement in the vertical direction (*y*-axis). The displacement was present in all ground elements when a cavity occurred in the numerically modeled ground and a load was applied to the top of the pavement. Here, the vertical displacement distribution is displayed in red when a large vertical displacement occurs. In addition, the cavity area was increased (0.79, 1.77, and 3.14 m^2^), the pavement thickness was increased (0.1 m and 0.3 m), the elastic modulus was decreased (3000 MPa and 500 MPa), and the traffic load was increased (12.7, 25.4, and 38.1 kN/m^2^), as shown according to the cavity type.

As shown in Figure 6, the vertical displacement at the top of the cavity increased with the increase in cavity area and traffic load and the decrease in pavement thickness and elastic modulus. According to this result, stress was released by nulling the part of the ground element modeled as a cavity. In addition, a large vertical displacement occurred because the loading on the pavement layer above the cavity had a great effect on the ground elements above the ground. Moreover, it was confirmed that the vertical displacement decreased as the distance from the center was gradually increased. This implies that the effect of stress release and loading was greatest at the center of the modeled ground and that this effect became smaller as the location was farther from the center. Therefore, the maximum surface settlement occurred at the top of the modeled ground center under all conditions. By comparing the results of Figure 6a,b and Figure 6c,d, it was confirmed that the vertical displacement occurring in the ground elements above the cavity was larger when the cavity was rectangular than that when it was circular. Moreover, by comparing the results of Figure 6a,c and Figure 6b,d, it was confirmed that the vertical displacement generated in the ground elements above the ground became larger when the pavement elastic modulus decreased. This could be also confirmed via the distribution chart trend. Figure 6c shows a case where the ground failure occurred before the analysis was completed under the conditions of a cavity area of 3.14 m^2^, a pavement thickness of 0.1 m, and traffic load of 38.1 kN/m^2^. This confirmed that the elastic modulus of the pavement layer had a direct effect on the ground where the cavity occurred.

When the cavity was rectangular, with increasing cavity area and traffic load and decreasing pavement thickness and elastic modulus, the vertical displacement of the ground elements above the cavity increased. Therefore, the overall reduction in ground stability where a cavity occurred according to the influencing factors could be confirmed by observing the vertical displacement distribution results.

### 3.2. Correlations between Cavity Type and Area and Surface Settlement

This section describes the correlations of the influencing factors analyzed using surface settlement, which is the result of a numerical analysis considering the factors affecting the ground above the cavity (cavity type and area, pavement thickness and elastic modulus, and load size). Herein, surface settlement means the maximum surface settlement that occurs during cavity development and loading at the top of the modeled ground center.

Figure 7, Figure 8 and Figure 9 show the surface settlement relationships according to the increase in cavity area for different values of pavement elastic modulus (3000, 1000, and 500 MPa), pavement thickness (0.1, 0.2, and 0.3 m), and traffic load (12.7, 25.4, and 38.1 kN/m^2^) for the two cavity types (rectangular and circular).

Figure 7 shows the surface settlement result for a pavement elastic modulus of 3000 MPa. When the cavity type was rectangular, the larger the cavity area and traffic load and the smaller the pavement thickness, the larger the surface settlement. This trend was the same under all analysis conditions. In particular, in the 0.1 m pavement thickness case in Figure 7c, where the surface settlement was the largest, when the cavity area increased from 0.79 m^2^ to 1.77 and 3.14 m^2^, the surface settlement increased by 51% and 150%, respectively, for a rectangular cavity, and by 36% and 91%, respectively, for a circular cavity. The same surface settlement trend was observed with a pavement elastic modulus of 1000 MPa, as shown in Figure 8. In the 0.1 m pavement thickness case in Figure 8c, where the surface settlement was the largest, when the cavity area was increased from 0.79 m^2^ to 1.77 and 3.14 m^2^, the surface settlement increased by 60% and 202%, respectively, for a rectangular cavity is rectangular, and by 38% and 98%, respectively, for a circular cavity. The surface settlement results with a pavement elastic modulus of 500 MPa, as shown in Figure 9, also showed a similar trend. In the 0.1 m pavement thickness case in Figure 9c, where the surface settlement was the largest, when the cavity area was increased from 0.79 m^2^ to 1.77 and 3.14 m^2^, the surface settlement increased by 67% and 254%, respectively, for a rectangular cavity, and by 38% and 100%, respectively, for a circular cavity.

From the analysis of the correlations between surface settlement and cavity type and area in Figure 7, Figure 8 and Figure 9, one can clearly observe differences in the surface settlement of the ground above a cavity according to the cavity type. This confirmed that the ground where a rectangular cavity occurred had a larger surface settlement. Moreover, the maximum surface settlement increased with the cavity area. The increase rate was larger when the cavity area was increased, the pavement thickness was decreased, and the cavity was rectangular. In addition, with a reduced elastic modulus of the pavement applied considering cracks in the pavement layer, the increase in the surface settlement with increasing cavity area was larger than with a large elastic modulus. In other words, the surface settlement had a large influence according to cavity type and area. Therefore, it should be considered an important factor affecting the ground above a cavity.

### 3.3. Correlations between Pavement Thickness, Elastic Modulus, and Surface Settlement

Figure 10, Figure 11 and Figure 12 show the surface settlement according to the increase in pavement thickness for different values of the pavement elastic modulus (3000, 1000, and 500 MPa), cavity area (0.79, 1.77, and 3.14 m^2^), and traffic load (12.7, 25.4, and 38.1 kN/m^2^), for the two cavity types (rectangular and circular).

Figure 10 shows the surface settlement result for a pavement elastic modulus of 3000 MPa. For the rectangular cavity, when the pavement thickness was decreased, the larger the cavity area and traffic load, the larger the surface settlement. This trend was the same under all analysis conditions. In particular, in the 3.14 m^2^ cavity area case in Figure 10c, where the surface settlement was the largest, when the pavement thickness was increased from 0.1 m to 0.2 and 0.3 m, the surface settlement decreased by 30% and 42%, respectively, for a rectangular cavity, and by 18% and 29%, respectively, for a circular cavity. The same surface settlement trend was observed with a pavement elastic modulus of 1000 MPa (Figure 11). In the 3.14 m^2^ cavity area case in Figure 11c, where the surface settlement was the largest, when the pavement thickness was increased from 0.1 m to 0.2 and 0.3 m, the surface settlement decreased by 33% and 45%, respectively, for a rectangular cavity, and by 13% and 23%, respectively, for a circular cavity. The surface settlement results with a pavement elastic modulus condition 500 MPa also showed a similar trend (Figure 12). In the 3.14 m^2^ cavity area case in Figure 12c, where the surface settlement was the largest, when the pavement thickness area was increased from 0.1 m to 0.2 and 0.3 m, the surface settlement decreased by 36% and 48%, respectively, for a rectangular cavity, and by 9% and 19%, respectively, for a circular cavity. 

From the analysis of the correlations between surface settlement, pavement thickness, and elastic modulus in Figure 10, Figure 11 and Figure 12, the differences in surface settlement decreased with increasing pavement thickness in the ground above a cavity. Therefore, the general effects of pavement thickness and elastic modulus on the ground above a cavity were identified. That is, for the largest pavement thickness, the surface settlement was small, and the reduction rate was the greatest. Furthermore, when the cavity area was large, the surface settlement reduction according to the increase in pavement thickness was large. This suggests that the cavity area had a significant effect on pavement thickness because the surface settlement was large even if the load was small when the cavity area was large. Moreover, even when the pavement elastic modulus was small, the surface settlement reduction according to the increase in pavement thickness was small. This also suggests that the reduction rate was small because the surface settlement was large when the pavement thickness was small. However, when the cavity type was rectangular and the cavity area and traffic load were the largest, the surface settlement reduction rate was the largest even when the pavement elastic modulus was the smallest. This suggests that the cavity area and traffic load had the largest influence on the surface settlement.

## 4. Conclusions

In this study, FEA was conducted considering various influencing factors (namely, cavity type and area, traffic load, pavement thickness, and elastic modulus) to evaluate the mechanical behavior of ground around a cavity. The results and findings of the analysis based on vertical displacement distribution and surface settlement can be summarized as follows.

(1)The results of the vertical displacement distribution showed that the largest amount of surface settlement occurred at the top of the model ground center. The vertical displacement decreased as the distance from the model ground center increased. This means that the effect of stress release and load was the greatest at the center of the model ground, and the effect became smaller farther from the center. For the rectangular cavity, it was qualitatively confirmed that the vertical displacement occurring in the ground element above the cavity was relatively larger than for a circular cavity. Therefore, it was found that the behavior of the ground around the cavity could be qualitatively evaluated from the results of the vertical displacement distribution.(2)When the elastic modulus was 3000 MPa under the condition of the smallest thickness of the pavement layer, the surface settlement of rectangular and circular cavities increased by 51% to 150% and by 36% to 91%, respectively, with the increase of the cavity size. This trend of increased surface settlement was the same even when the elastic modulus was 1000 MPa and 500 MPa. This means that the surface settlement increased as the cavity size increased, and the circular cavity was more stable than the rectangular cavity.(3)When the cavity size was the largest under the condition of the smallest thickness of the pavement, the surface settlement of the rectangular and circular cavities decreased by 30% to 42% and by 18% to 29%, respectively, with the increase of the cavity thickness. The surface settlement decreased as the elastic modulus decreased.(4)Based on the FEA results, the mechanical behavior of ground around a cavity was successfully evaluated considering influencing factors such as cavity type and area, traffic load, pavement thickness, and elastic modulus. Furthermore, the degree of stability decrease underground where a cavity occurred could be evaluated through the correlations between each influencing factor and surface settlement. However, the stability should be quantitatively evaluated, and predictive indicators should be suggested in future work considering more variable analysis conditions. It is also necessary to study other conditions of the ground, because this study reports the numerical analysis results of limited conditions.

## Figures and Tables

**Figure 1 materials-15-08301-f001:**
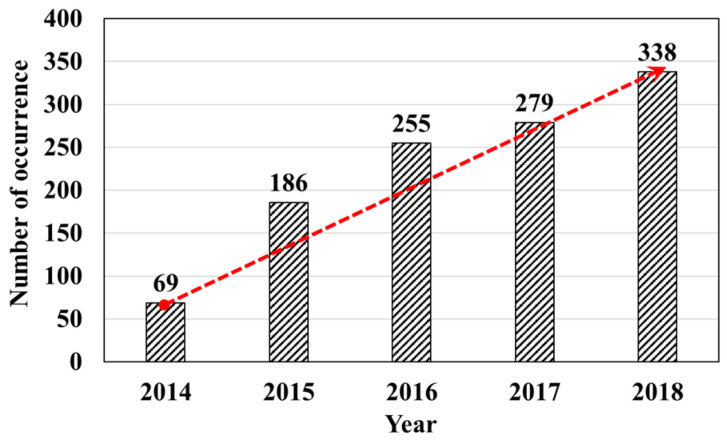
Occurrence frequency of ground subsidence episodes from 2014 to 2018 in Korea.

**Figure 2 materials-15-08301-f002:**
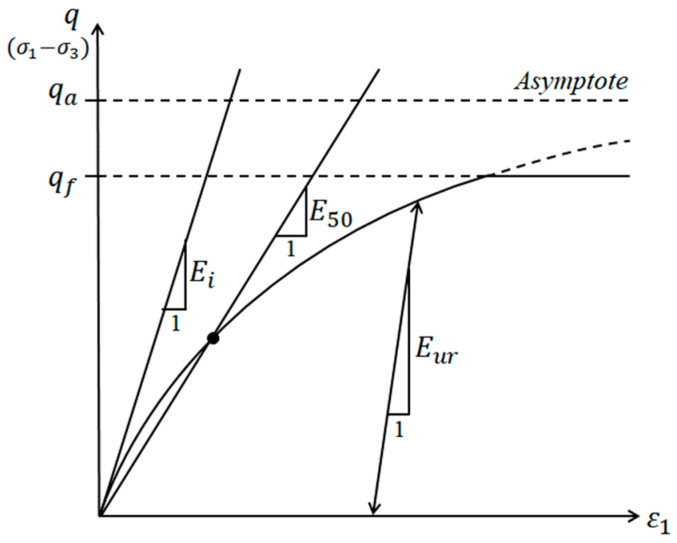
Relationship between stress and strain in the hardening soil model [39].

**Figure 3 materials-15-08301-f003:**
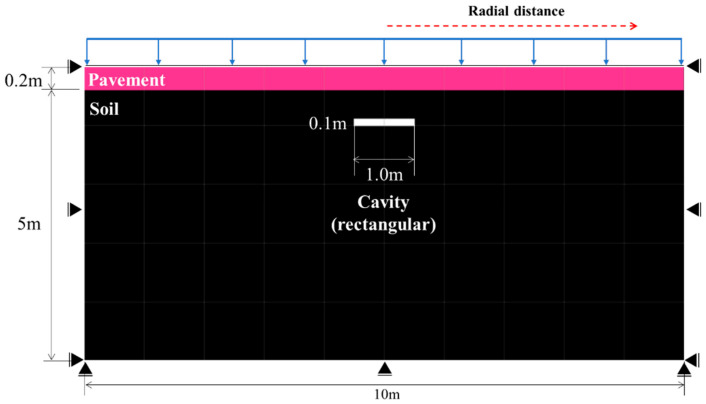
Numerical analysis model.

**Figure 4 materials-15-08301-f004:**
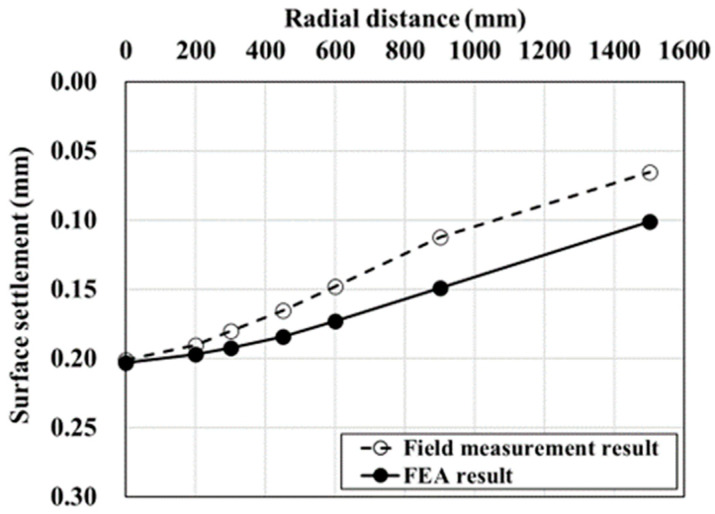
Numerical analysis verification results.

**Figure 5 materials-15-08301-f005:**
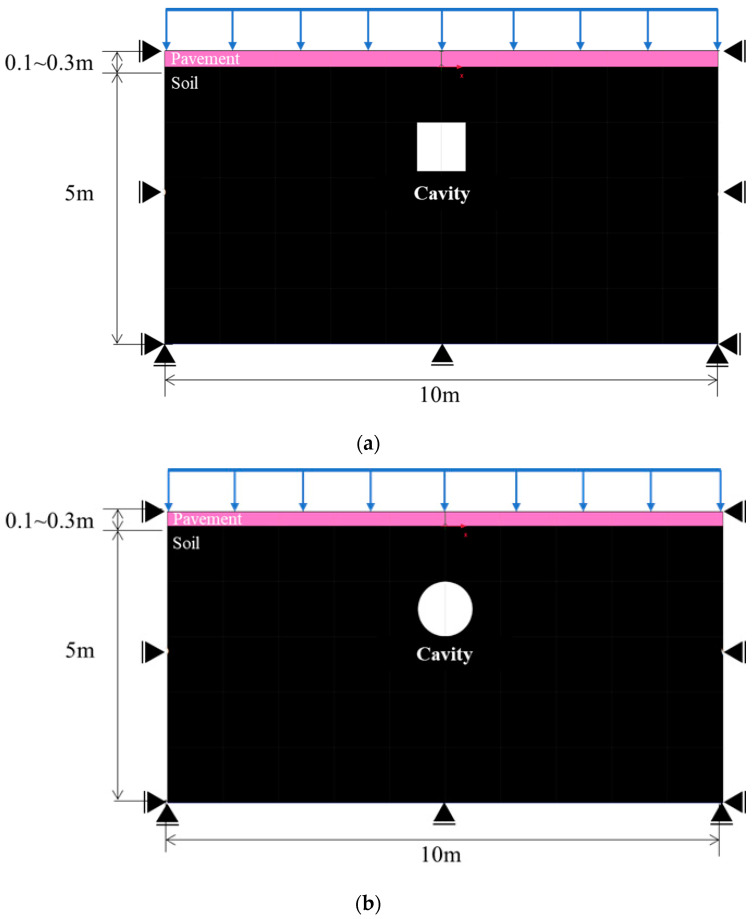
Numerical analysis model for the evaluation of influencing factors. (**a**) Rectangular and (**b**) circular cavities.

**Figure 6 materials-15-08301-f006:**
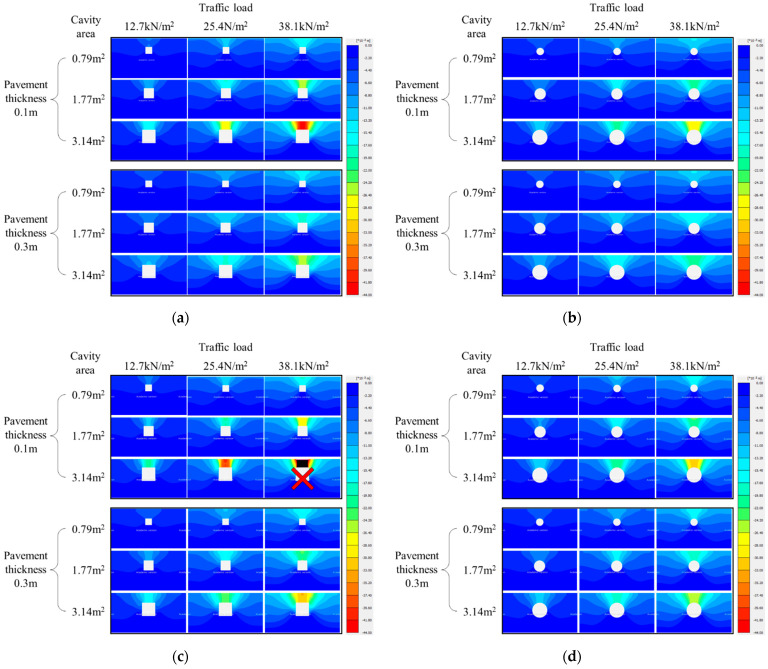
Vertical displacement distribution. (**a**) Rectangular cavity (3000 MPa); (**b**) circular cavity (3000 MPa); (**c**) rectangular cavity (500 MPa); (**d**) circular cavity (500 MPa).

**Figure 7 materials-15-08301-f007:**
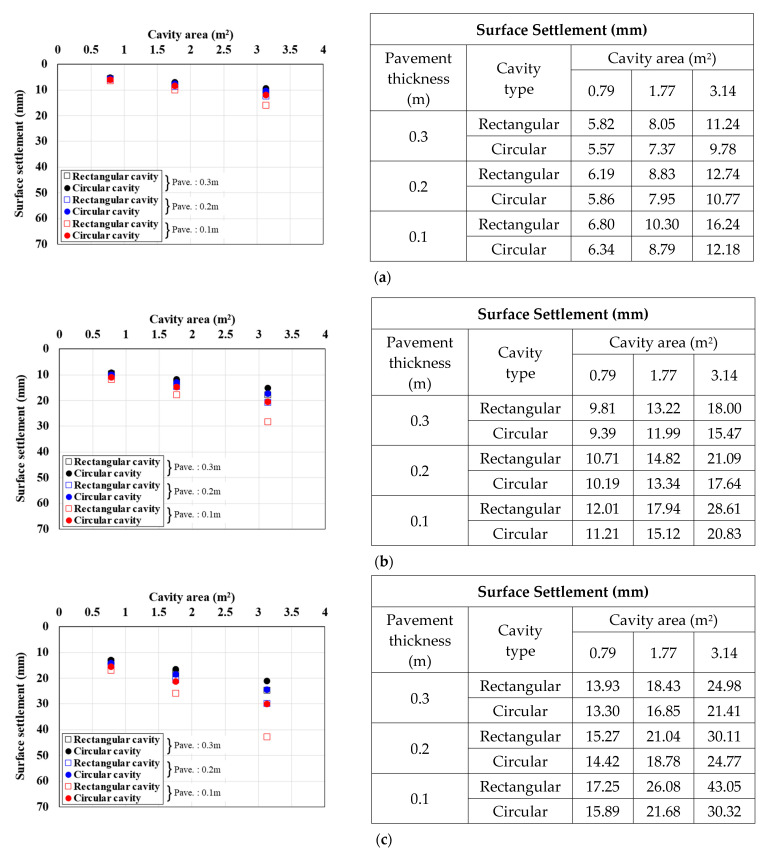
Surface settlement according to cavity area by pavement thickness (pavement elastic modulus: 3000 MPa). Traffic load: (**a**) 12.7, (**b**) 25.4, and (**c**) 38.1 kN/m^2^.

**Figure 8 materials-15-08301-f008:**
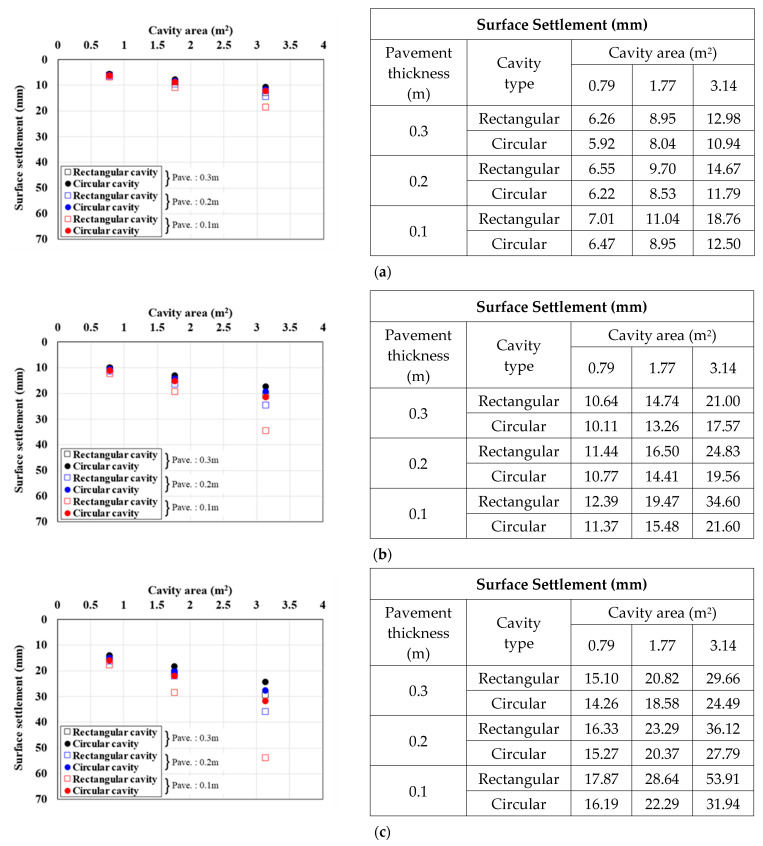
Surface settlement according to cavity area by pavement thickness (pavement elastic modulus: 1000 MPa). Traffic load: (**a**) 12.7, (**b**) 25.4, and (**c**) 38.1 kN/m^2^.

**Figure 9 materials-15-08301-f009:**
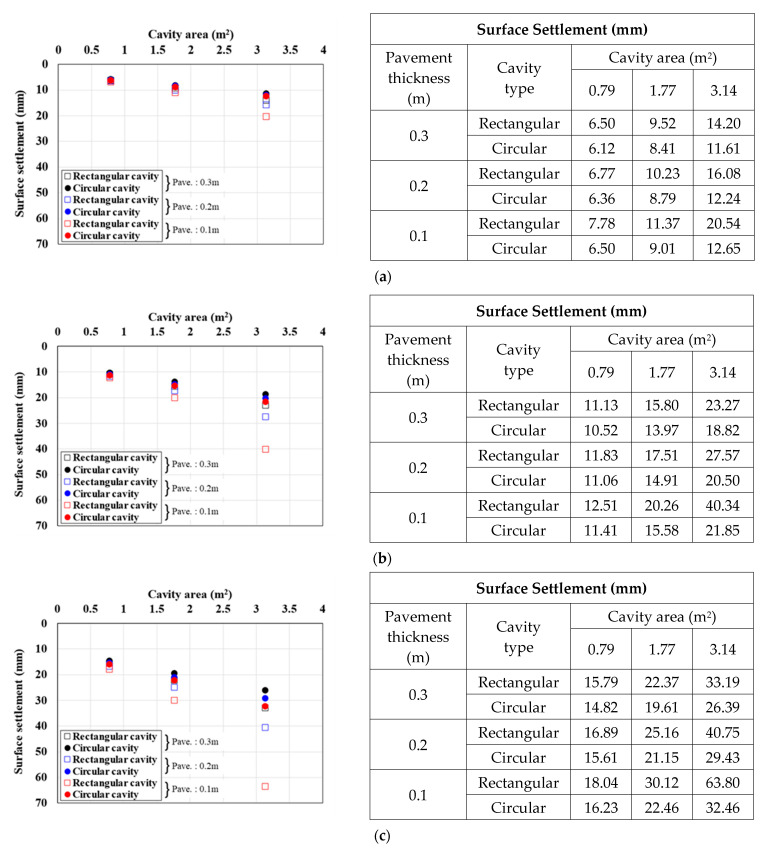
Surface settlement according to cavity area by pavement thickness (pavement elastic modulus: 500 MPa). Traffic load: (**a**) 12.7, (**b**) 25.4, and (**c**) 38.1 kN/m^2^.

**Figure 10 materials-15-08301-f010:**
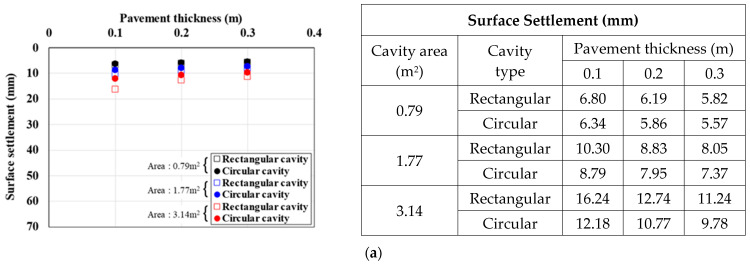
Surface settlement according to pavement thickness by cavity area (pavement elastic modulus: 3000 MPa). Traffic load: (**a**) 12.7, (**b**) 25.4, and (**c**) 38.1.

**Figure 11 materials-15-08301-f011:**
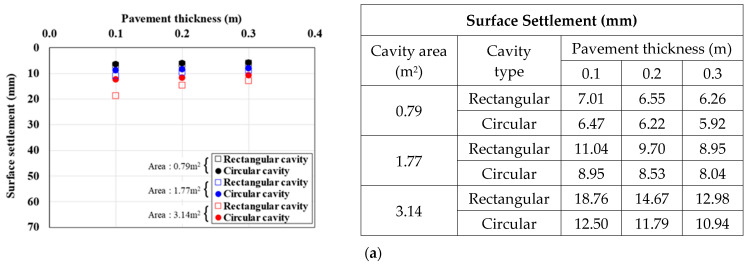
Surface settlement according to pavement thickness by cavity area (pavement elastic modulus: 1000 MPa). Traffic load: (**a**) 12.7, (**b**) 25.4, and (**c**) 38.1.

**Figure 12 materials-15-08301-f012:**
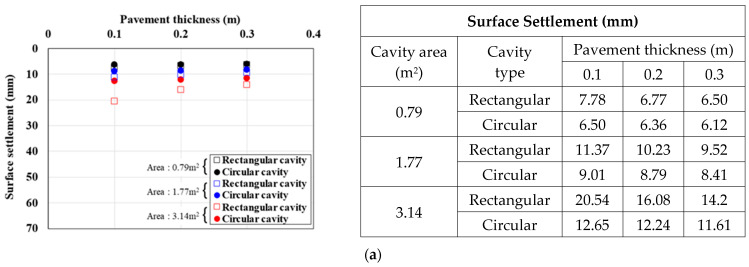
Surface settlement according to pavement thickness by cavity area (pavement elastic modulus: 500 MPa). Traffic load: (**a**) 12.7, (**b**) 25.4, and (**c**) 38.1.

**Table 1 materials-15-08301-t001:** Soil parameters of the hardening soil model in the numerical analysis [30].

$E_{50}^{\mathrm{ref}}\;\;(\mathrm{kPa})$	$E_{\mathrm{oed}}^{\mathrm{ref}}\;\;(\mathrm{kPa})$	$E_{\mathrm{ur}}^{\mathrm{ref}}\;\;(\mathrm{kPa})$	$c^{\prime }\;\;(\mathrm{kPa})$	$\varphi^{\prime }\;\;({}^{\circ})$	ψ (°)	γ (kN/m^3^)
16,000	22,000	90,000	0.5	37	1.5	18.0

**Table 2 materials-15-08301-t002:** Pavement parameters in the numerical analysis.

E (Mpa)	ν	γ (kN/m^3^)
3000	0.35	24.0

**Table 3 materials-15-08301-t003:** Numerical analysis cases.

Cavity Type	Cavity Area (m^2^)	Pavement Thickness (m)	Pavement Elastic Modulus (MPa)	Traffic Load (kN/m^2^)
Rectangular & Circular	0.79	0.1, 0.2, 0.3	500, 1000, 3000	12.7
				25.4
				38.1
	1.77	0.1, 0.2, 0.3	500, 1000, 3000	12.7
				25.4
				38.1
	3.14	0.1, 0.2, 0.3	500, 1000, 3000	12.7
				25.4
				38.1

## Data Availability

Not applicable.

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
