# Peer review of "Numerical Analysis of Factors Influencing the Ground Surface Settlement above a Cavity"

_materials, 2022, doi:10.3390/ma15238301_

Round 1

Reviewer 1 Report

This paper presents the evaluation through vertical displacement distribution and surface settlement, and a finite element analysis was employed to ananlyze  the reliability of the road underground with a cavity under some influence factors, such as such as cavity area, pavement layer, and surcharge,  especially the cavity area and shape; but some problems are as following: (1) how to consider the interfacial effect between soil and pavement in the numerical model; (2) how to consider the confining stress for the cavity? (3) Conclusion1 can not be as a conclusion, and other conclusions are not enough clearly; (4) this paper overall is relatively simple, and numerical model is also simply, such as, all well known, the circular cavity is superior to rectangular cavity; the correlation of influence factors vague.

Reviewer 2 Report

minor revision

Reviewer 3 Report

The manuscript presents a study on the influence of cavity type and area, traffic load, pavement thickness, and elastic modulus on the surface settlements of a road. In general, the research presents very interesting findings, although the approach and methods are found to be sophisticated. The scientific value and soundness could be improved by focusing on important details, like soil parameters, numerical models description, and investigation techniques. The manuscript is well-written and structured with no major remarks concerning the approach, results or conclusions. The comments mostly refer to material properties used, and the presentation of the results that could be improved. The abstract and introduction parts need a clear formulated aim and the objectives of the study. Furthermore, the contribution of the research needs to be clearly presented. What is the advantage of the present study and what distinguishes it from others? Some of the specific comments can be followed below: l.22 ground stability needs to be clearly defined. do the authors refer to any of the limit states, ULS, SLS? fig. 1 Are the results presented on a global scale or only in S. Korea? Please specify in the caption l.108 The introduction should include a description of the subsidence mechanism in detail. Explaining the entire process. l.132 more information on the soil material is needed. Since subsidence is the case then hydraulic parameters or suffusion should be related. tab. 1 Please explain where this data comes from. Is it a particular case study? How the data was obtained, from the lab or in situ tests? l.138 (tab.1) please be specific, are these effective values? if so then marked the cohesion with prime and provide information on the pore water pressures fig.2 This figure does not present any information related to the parameters. Please explain all the parameters used in the modelling l.142 More focus on site investigation needs to be placed. Please provide details on the survey l.164 This is no surprise, the established models are very much accurate. Please expose the real contribution here. fig.3 Please explain the modelled (chosen) size and the shape of the cavity area. fig.4 Please comment on how significant are such settlements in practice and how they influence road performance l.190 The vertical displacement needs to be present in real numbers, the legend in fig 6 misses the scale and units. Please be more accurate and specific by showing the values for settlements. fig. 6 Please think of another way of presenting the results. The present form is unclear and confusing. Is the lower part where the thickness was increased? The values of increased traffic, area, and pavement should be provided. fig 7-12. the scale of surf. the settlement could be increased for better presentation and understanding   These are only the specific comments, more remarks (typos, formatting, units) can be found in the marked copy of the manuscript.  
